# TestDG: Test-time Domain Generalization for Continual Test-time Adaptation

## Abstract

This paper studies continual test-time adaptation (CTTA), the task of adapting a model to constantly changing unseen domains in testing while preserving previously learned knowledge. Existing CTTA methods mostly focus on adaptation to the current test domain only, overlooking generalization to arbitrary test domains a model may face in the future. To tackle this limitation, we present a novel online test-time domain generalization framework for CTTA, dubbed TestDG. TestDG aims to learn features invariant to both current and previous test domains on the fly during testing, improving the potential for effective generalization to future domains. To this end, we propose a new model architecture and a test-time adaptation strategy dedicated to learning domain-invariant features, along with a new data structure and optimization algorithm for effectively managing information from previous test domains. TestDG achieved state of the art on four public CTTA benchmarks. Moreover, it showed superior generalization to unseen test domains.

## 1 Introduction

Deep neural networks have driven significant advances in various vision tasks such as classification (He et al., 2016; Li et al., 2019; Dosovitskiy et al., 2021), object detection (Girshick et al., 2014; Carion et al., 2020; Cai & Vasconcelos, 2018), and semantic segmentation (Noh et al., 2015; Xie et al., 2021; Chen et al., 2018). Despite these achievements, they often struggle with limited generalization ability to the domain shift between training and test data (Ganin & Lempitsky, 2015; Hendrycks & Dietterich, 2019; Sun & Saenko, 2016; Chang et al., 2019). Test-time adaptation (TTA) (Zhao et al., 2023; Wang et al., 2021a; Zhang et al., 2022; Niu et al., 2022; 2023) has been developed to mitigate the distribution shift by adapting a pre-trained model to unlabeled test domains during testing. In specific, TTA methods update the model in testing on the fly, by self-training using pseudo-labeled test data (Song et al., 2023; Gan et al., 2023; Liu et al., 2024b) or by entropy minimization of the model prediction (Niu et al., 2022; Wang et al., 2021a). Early approaches to TTA assume a single and fixed test domain. In reality, however, this assumption does not hold since a model is often deployed in non-stationary and continually changing environments. In such environments, the accuracy of the model can easily degrade even with TTA since the constant distribution shift in testing causes unreliable pseudo labels (Guo et al., 2017), which in turn exacerbates error accumulation and catastrophic forgetting.

Continual test-time adaptation (CTTA) (Wang et al., 2022; Song et al., 2023; Gan et al., 2023; Liu et al., 2024b; Lee et al., 2024; Yang et al., 2024a; Liu et al., 2024a; Yang et al., 2024b; Zhu et al., 2024) has been studied to address these realistic TTA scenarios. Most of existing CTTA methods focus only on adapting a model to the test domain at hand. Although these methods lead to notable performance improvements, there remains room for further improvement in that they do not fully account for the model's generalization to future test domains it encounters later in continuously changing environments.

In this paper, we present an online TEST-time Domain Generalization framework for continual test-time adaptation, dubbed TestDG. TestDG overcomes the aforementioned limitation by learning domain-invariant representations that well generalize to unseen domains on the fly during testing. This approach improves robustness of a model to arbitrary test domains that it may encounter in the future, as demonstrated in

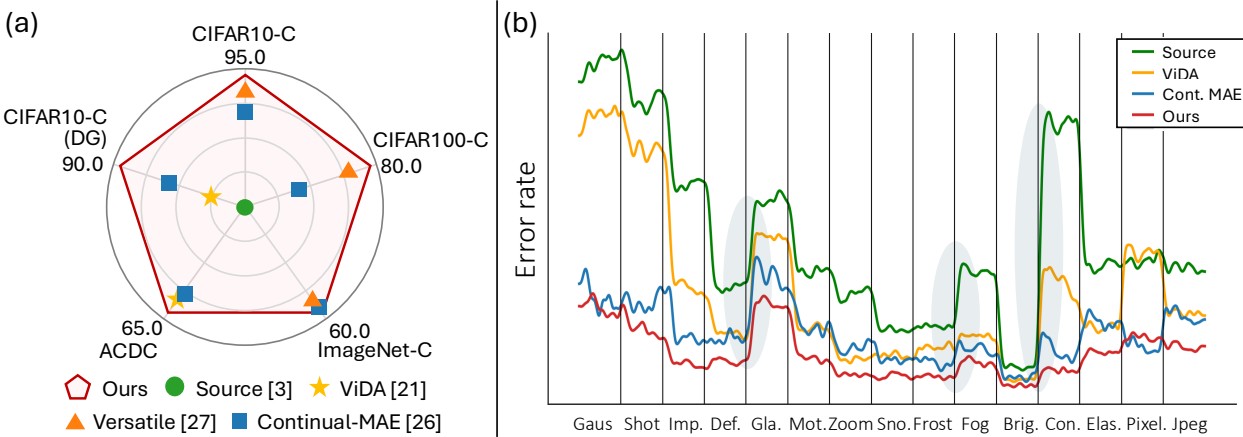

Figure 1: Empirical demonstration of the effectiveness of TestDG. (a) Accuracy across four CTTA and one domain generalization (DG) benchmarks, where DG performance is evaluated on 5 unseen domains after CTTA over 10 domains. For ACDC, mIoU is used instead. *TestDG achieved the best on all benchmarks.* (b) Error rate over sequentially changing test domains. The consistently low error rates and notably stable performance of TestDG, especially in moments of domain change, highlight its robustness.

Figure 1(a). Furthermore, as shown in Figure 1(b), learned domain-generalizable features enable robust adaptation even under abrupt domain shifts. A naïve approach to implementing domain-invariant learning is to align features from different test domains. However, since the features contain their own semantic contents as well as domain-specific information, this alignment without consideration of semantics corrupts the features and consequently degrades performance.

Hence, we introduce *domain information extractor* that takes the encoder features as input and extracts only domain-specific information in the form of embedding vectors; we call such vectors *domain embeddings*. The encoder is, in turn, trained to make domain embeddings from different domains indistinguishable by reducing the gap between such domain embeddings, so that it learns domain-invariant representation. Analogous to adversarial learning (Goodfellow et al., 2020; Ganin & Lempitsky, 2015; Li et al., 2018), we alternate the optimization of the domain information extractor and that of the encoder. This approach allows the encoder to gradually become domain-invariant, and the domain information extractor to become sensitive to the remaining domain-specific information of the encoder.

The remaining challenge in implementing TestDG is that only the test domain at hand is accessible at each iteration of CTTA, while domain-invariant learning demands domain embeddings of at least two test domains. To tackle this, TestDG efficiently stores embedding vectors that represent domain embeddings of the previous test domain; we call such embedding vectors *domain prototypes* and sample them from domain embeddings of the previous domain through submodular optimization (Nemhauser et al., 1978; Kim et al., 2016). By encouraging that domain embeddings of the current domain are not distinguished from the domain prototypes of the previous domain, TestDG performs domain-invariant learning of the encoder without access to samples of previous test domains.

TestDG was evaluated on the four public benchmarks for CTTA: CIFAR10-to-CIFAR10-C (Krizhevsky et al., 2009; Hendrycks & Dietterich, 2019), CIFAR100-to-CIFAR100-C (Krizhevsky et al., 2009; Hendrycks & Dietterich, 2019), ImageNet-to-ImageNet-C (Deng et al., 2009; Hendrycks & Dietterich, 2019), and Cityscapes-to-ACDC (Cordts et al., 2016; Sakaridis et al., 2021). It achieved a new state of the art on all the benchmarks and demonstrated greater generalization ability on unseen test domains.

## 2   Related work

**Test-time adaptation** Recent research on TTA aims at addressing the domain shift problem by adapting a pre-trained model online during testing (Zhao et al., 2023; Sun et al., 2020; Wang et al., 2021a; 2022; Liu et al.,

2024b; Zhang et al., 2022). Initial studies (Sun et al., 2020; Liu et al., 2021) employed self-supervised learning with proxy tasks on unlabeled test data, but they require modifications to the training stage to accommodate the proxy tasks. Meanwhile, Tent (Wang et al., 2021a), a fully test-time adaptation method based on entropy minimization, requires no modifications to the training stage and is applicable to any pre-trained model. Building on this work, several studies (Niu et al., 2022; 2023) have proposed using entropy-based losses in testing environments. Other approaches (Su et al., 2022; Liu et al., 2021; Eastwood et al., 2021) leveraged lightweight information from source data to facilitate adaptation while mitigating catastrophic forgetting. Unlike them, our method directly extracts domain embeddings from streamed test data, without computation for source data.

**Continual test-time adaptation** In real-world scenarios, testing environments can vary significantly due to multiple factors such as weather, time, and geolocation. Since conventional TTA methods have been studied under the assumption that the test domain is static, they face challenges in continually changing environments. To address this limitation, Wang et al. (2022) introduced the first CTTA method, leveraging multiple augmented inputs and a momentum network to generate more reliable pseudo labels. Song et al. (2023) and Lee et al. (2024) proposed memory-efficient methods that utilized lightweight meta networks and a mixture of experts, respectively. Gan et al. (2023) and Liu et al. (2024b) utilized visual prompt learning and visual domain adapters, respectively, to disentangle domain-specific and domain-shared knowledge. Liu et al. (2024a) improved target domain knowledge extraction through a self-supervised framework. In this context, Yang et al. (2024b) proposed a versatile network that identifies and refines unreliable pseudo-labels using the source pre-trained model, while Zhu et al. (2024) introduced an uncertainty-aware buffering and graph-based constraint. Unlike these methods, TestDG learns domain-invariant features that well generalize to diverse unseen test domains.

**Domain generalization** Domain generalization (DG) focuses on improving the generalization performance of models on unseen target domains by training them with multiple source domains. Initial work in DG employed domain alignment (Li et al., 2018; Shao et al., 2019; Jia et al., 2020; Li et al., 2020; Wang et al., 2021b) to learn domain-invariant representations by minimizing the distance of distribution among multiple domains. Recent studies in DG have introduced various strategies, including data augmentation (Zhou et al., 2020a;b), meta learning (Balaji et al., 2018; Dou et al., 2019), and representation learning (Nam et al., 2021; Harary et al., 2022). Inspired by DG, this paper presents a CTTA method that updates an online model through domain-invariant learning to handle continually changing domains.

## 3 Proposed method

TestDG is an online test-time domain generalization framework for CTTA, designed to learn domain-invariant features from a stream of test-time inputs; its overall pipeline is illustrated in Figure 2. Unlike domain adaptation and generalization, which in general assume that data from at least two domains are available during the entire training process, CTTA allows access to a handful of data from the test domain at hand (*i.e.*, the input data at each test point) for each iteration. This constraint poses a non-trivial challenge for domain-invariant learning.

TestDG addresses this challenge by storing *domain embeddings*, *i.e.*, embedding vectors containing domain information of the most recent previous domain, in a queue. To prevent excessive memory consumption caused by storing all domain embeddings, we select representative embeddings from the queue of domain embeddings as domain prototypes when the test domain changes.

To achieve domain-invariant learning without corrupting the semantics of features, TestDG operates in two steps as follows. In the first step, the domain information extractor learns to extract domain embeddings, by capturing only domain-specific information from the encoder features. To facilitate this extraction, another extra module called *domain amplifier* is attached to the encoder for capturing and emphasizing the domain-specific information of the encoder features. We design the domain amplifier as a lightweight adapter structure (Chen et al., 2022; Houlsby et al., 2019; Pfeiffer et al., 2020) to minimize additional parameters. During this step, the domain amplifier is updated alongside the domain information extractor while the encoder remains frozen. In the second step, we perform domain-invariant learning by ensuring the domain embeddings of the current domain become indistinguishable from the domain prototypes of the previous

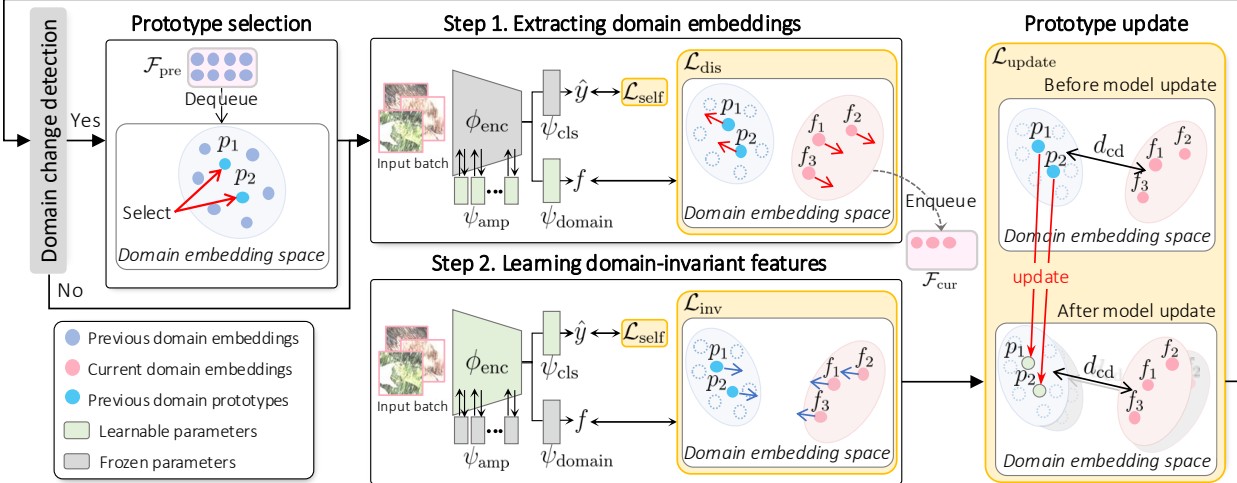

Figure 2: Overview of TestDG. When a domain change is detected, TestDG selects domain prototypes that well represent the previous domain embeddings stored in a queue. Each iteration consists of two steps. In Step 1, the domain information extractor $\psi_{\mathrm{domain}}$ learns alongside domain amplifier $\psi_{\mathrm{amp}}$ to extract domain embeddings from the encoder features using the domain discrimination loss $\mathcal{L}_{\mathrm{dis}}$. In Step 2, the encoder $\phi_{\mathrm{enc}}$ is trained using the domain-invariant loss $\mathcal{L}_{\mathrm{inv}}$ to reduce the gap between domain embeddings and domain prototypes of the previous domain. The domain prototypes are continuously updated to match with the updated model parameters by $\mathcal{L}_{\mathrm{update}}$.

domain. During this step, only the encoder is updated, with the domain information extractor and domain amplifier frozen.

By alternating between the two steps, the encoder gradually learns domain-invariant features across multiple test domains. Meanwhile, the domain information extractor and domain amplifier are updated to reflect the update of the encoder and extract domain embeddings corresponding to the current state of the encoder. In addition, TestDG continuously updates the domain prototypes to match them with the updated model parameters. Since the model used to extract domain prototypes is constantly updated during CTTA, the prototypes must also be updated accordingly.

## 3.1 Domain change detection

As it is unknown when the test domain changes in CTTA, we detect domain changes based on prediction confidence scores as in Gan et al. (2023). That is, if the difference between confidence scores of two consecutive predictions is greater than a threshold, the test domain is considered to have changed. TestDG is designed to remain effective even with detection errors: false positives are caused by intra-domain shifts and thus let TestDG learn invariant features within a single domain, while false negatives suggest mild distribution shifts that can be handled robustly by the model at hand as-is.

## 3.2 Domain prototype selection

When a domain change is detected, TestDG retains a small set of $n$ domain prototypes $\mathcal{P}_{\mathrm{pre}} = \{p_i\}_{i=1}^{n}$ from the previous domain embeddings in the queue $\mathcal{F}_{\mathrm{pre}} = \{f_i\}_{i=1}^{m}$ to reduce memory footprint substantially. To ensure that these prototypes represent the entire distribution of the previous domain, we select $\mathcal{P}_{\mathrm{pre}}$ by minimizing the discrepancy between $\mathcal{F}_{\mathrm{pre}}$ and $\mathcal{P}_{\mathrm{pre}}$. Following the prototype sampling technique (Kim et al., 2016), we measure the discrepancy between $\mathcal{F}_{\mathrm{pre}}$ and $\mathcal{P}_{\mathrm{pre}}$ using the squared maximum mean discrepancy

(MMD):

$$
\begin{aligned}
\mathrm{MMD}^2(\mathcal{F}_{\mathrm{pre}}, \mathcal{P}_{\mathrm{pre}}) := {} & \frac{1}{|\mathcal{F}_{\mathrm{pre}}|^2} \sum_{f_i, f_j \in \mathcal{F}_{\mathrm{pre}}} k(f_i, f_j) \\
& - \frac{2}{|\mathcal{F}_{\mathrm{pre}}||\mathcal{P}_{\mathrm{pre}}|} \sum_{f_i \in \mathcal{F}_{\mathrm{pre}}, p_j \in \mathcal{P}_{\mathrm{pre}}} k(f_i, p_j) \\
& + \frac{1}{|\mathcal{P}_{\mathrm{pre}}|^2} \sum_{p_i, p_j \in \mathcal{P}_{\mathrm{pre}}} k(p_i, p_j) \,,
\end{aligned}
\tag{1}
$$

where $k(\mathbf{x}, \mathbf{x}') = \exp(-\gamma ||\mathbf{x} - \mathbf{x}'||^2)$ is a RBF kernel that measures the discrepancy between embeddings from the two sets. As the first term in Eq. (1) remains constant with respect to $\mathcal{P}_{\mathrm{pre}}$, we define a score function $J(\mathcal{P}_{\mathrm{pre}})$ as follows:

$$
\begin{aligned}
J(\mathcal{P}_{\mathrm{pre}}) := {} & \mathrm{MMD}^2(\mathcal{F}_{\mathrm{pre}}, \emptyset) - \mathrm{MMD}^2(\mathcal{F}_{\mathrm{pre}}, \mathcal{P}_{\mathrm{pre}}) \\
= {} & \frac{2}{|\mathcal{F}_{\mathrm{pre}}||\mathcal{P}_{\mathrm{pre}}|} \sum_{f_i \in \mathcal{F}_{\mathrm{pre}}, p_j \in \mathcal{P}_{\mathrm{pre}}} k(f_i, p_j) \\
& - \frac{1}{|\mathcal{P}_{\mathrm{pre}}|^2} \sum_{p_i, p_j \in \mathcal{P}_{\mathrm{pre}}} k(p_i, p_j) \,,
\end{aligned}
\tag{2}
$$

where the constant term $\mathrm{MMD}^2(\mathcal{F}_{\mathrm{pre}}, \emptyset)$ is added to ensure that $J(\emptyset) = 0$. The domain prototypes are selected by maximizing the score function:

$$
\max_{\mathcal{P}_{\mathrm{pre}} \subset \mathcal{F}_{\mathrm{pre}} : |\mathcal{P}_{\mathrm{pre}}| = n} J(\mathcal{P}_{\mathrm{pre}}) \,.
\tag{3}
$$

This optimization is intractable, but a near-optimal solution can be achieved through a greedy process since the function $J$ is normalized monotone submodular (Nemhauser et al., 1978) when using the RBF kernel for $k(\cdot, \cdot)$, as proved in Kim et al. (2016). By maximizing the score function $J(\mathcal{P}_{\mathrm{pre}})$, selected prototypes closely approximate the entire distribution of domain embeddings in the queue. At the initial stage where no previous test domain exists, we use domain embeddings of augmented test images as domain prototypes.

### 3.3 Domain-invariant learning

TestDG updates the model through domain-invariant learning and self-training. For domain-invariant learning, we employ a two-step process. In the first step, the domain information extractor and amplifier are updated to extract domain-specific information, *i.e.*, domain embeddings, from the encoder features. In the second step, the encoder is updated to be domain-invariant by aligning the current domain embeddings with the previous domain prototypes (Sec. 3.2). Alternating these two steps allows the encoder to gradually become domain-invariant, and the domain information extractor and amplifier to become sensitive to the remaining domain-specific information of the encoder. In addition, following previous work (Wang et al., 2022; Gan et al., 2023; Liu et al., 2024b), we update our model by a self-training loss $\mathcal{L}_{\mathrm{self}}$, a cross-entropy loss between our model's prediction $\hat{y}$ and a pseudo label $\tilde{y}$ generated by its exponential moving average (Tarvainen & Valpola, 2017):

$$
\mathcal{L}_{\mathrm{self}}(x) = -\frac{1}{C} \sum_{c=1}^{C} \tilde{y}_c \log \hat{y}_c,
\tag{4}
$$

where $C$ is the total number of classes. For inference, we use $\hat{y}$ as the classification prediction. Details of the two steps are described below.

**Step 1: Extracting domain embeddings.** To maintain the semantic information of the encoder features during domain-invariant learning, we first extract their domain-specific information in the form of embedding vectors. To this end, we introduce the domain information extractor that extracts domain embeddings from

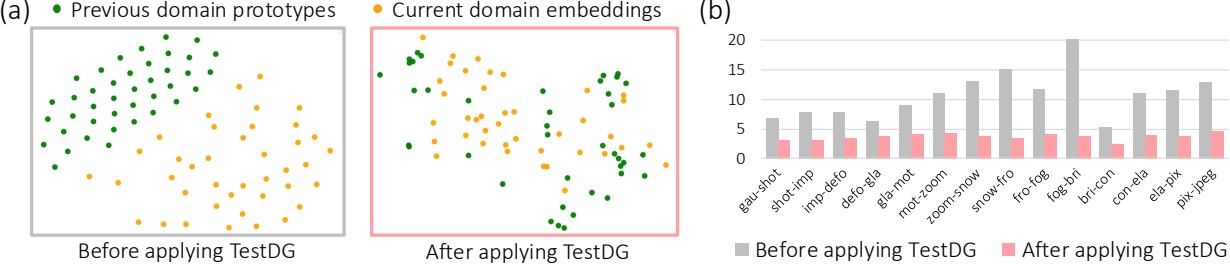

Figure 3: Analysis on the impact of domain-invariant learning in TestDG. (a) 2D visualization of the distribution of domain embeddings from different domains before and after TestDG. (b) The domain gap between different domains measured by Chamfer distance before and after applying TestDG.

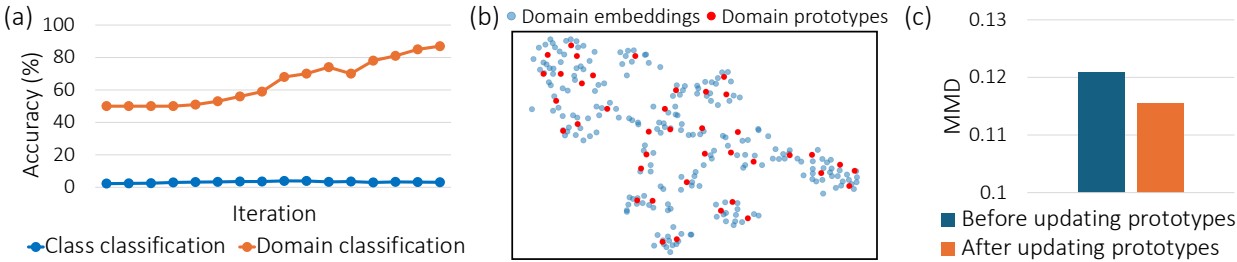

Figure 4: Analysis on domain embeddings and prototypes. (a) Analysis of class classification and domain classification accuracy using domain embeddings during CTTA iterations. (b) 2D visualization of domain embeddings and domain prototypes. (c) MMD between the target and original prototypes before the update and that between the target and updated prototypes.

output features of the encoder, and the domain amplifier that amplifies the domain-specific information of the encoder to facilitate the extractor. Let $\mathcal{F}_{\text{cur}} = \{f_i\}_{i=1}^n$ be the domain embeddings of test input images from the current test domain and $\mathcal{P}_{\text{pre}}$ the previous domain's prototypes. The domain information extractor and amplifier learn a domain embedding space by distinguishing $\mathcal{F}_{\text{cur}}$ and $\mathcal{P}_{\text{pre}}$ with a domain discrimination loss:

$$\mathcal{L}_{\text{dis}}(\mathcal{F}_{\text{cur}}, \mathcal{P}_{\text{pre}}) = -\frac{1}{n}\Bigg\{ \sum_{f_i \in \mathcal{F}_{\text{cur}},} \log(D(f_i))$$
$$- \sum_{p_i \in \mathcal{P}_{\text{pre}}} \log(1 - D(p_i))\Bigg\}, \qquad (5)$$

where $D(\cdot)$ indicates a binary linear classifier that serves as a domain discriminator. Minimizing this loss encourages the domain extractor with the amplifier to learn a space where embeddings of different domains are separated and thus capture domain-specific information.

**Step 2: Learning domain-invariant features.** The second step aims to optimize the encoder to achieve domain invariance across diverse test domains alongside classification (Eq. 4). TestDG accomplishes this by aligning current domain embeddings and previous domain prototypes. To be specific, we update the encoder and its classification head while freezing the domain information extractor and amplifier. Given current domain embeddings $\mathcal{F}_{\text{cur}}$ and previous domain prototypes $\mathcal{P}_{\text{pre}}$, the domain-invariant learning loss is given by

$$\mathcal{L}_{\text{inv}}(\mathcal{F}_{\text{cur}}, \mathcal{P}_{\text{pre}}) = \frac{1}{n} \sum_{(f_i, p_i) \in (\mathcal{F}_{\text{cur}}, \mathcal{P}_{\text{pre}})} \|f_i - p_i\|_1 . \qquad (6)$$

By minimizing this loss, the encoder's features are trained to produce domain embeddings that are indistinguishable between different domains, making them domain-generalizable to future domains.

### 3.4 Domain prototype update

Since the model used to extract the domain prototypes is constantly updated during CTTA, the domain prototypes extracted from the earlier model become outdated. Thus, the domain prototypes should also be updated to reflect the update of the model, approximating prototypes extracted from the updated model. Inspired by Asadi et al. (2023), we aim to preserve the set similarity between the previous domain prototypes $\mathcal{P}_{\mathrm{pre}}$ and the current domain embeddings $\mathcal{F}_{\mathrm{cur}}$ to effectively keep the relationship between them intact. In specific, we measure the Chamfer distance between the two sets $\mathcal{P}_{\mathrm{pre}}$ and $\mathcal{F}_{\mathrm{cur}}$. For iteration $t$, we update the domain prototypes $\mathcal{P}_{\mathrm{pre}}^{t-1}$ to $\mathcal{P}_{\mathrm{pre}}^{t}$ to ensure that the Chamfer distance between previous domain prototypes and current domain embeddings remains consistent before and after the model update. Specifically, the domain prototypes $\mathcal{P}_{\mathrm{pre}}^{t}$ are updated by minimizing the prototype updating loss:

$$\mathcal{L}_{\mathrm{update}} = |d_{\mathrm{CD}}(\mathcal{F}_{\mathrm{cur}}^{t-1}, \mathcal{P}_{\mathrm{pre}}^{t-1}) - d_{\mathrm{CD}}(\mathcal{F}_{\mathrm{cur}}^{t}, \mathcal{P}_{\mathrm{pre}}^{t})|, \tag{7}$$

where the Chamfer distance $d_{\mathrm{CD}}$ is computed by

$$d_{\mathrm{CD}}(\mathcal{F}_{\mathrm{cur}}, \mathcal{P}_{\mathrm{pre}}) = \sum_{f_i \in \mathcal{F}_{\mathrm{cur}}} \min_{p_j \in \mathcal{P}_{\mathrm{pre}}} \|p_j - f_i\|_2^2$$
$$+ \sum_{p_j \in \mathcal{P}_{\mathrm{pre}}} \min_{f_i \in \mathcal{F}_{\mathrm{cur}}} \|p_j - f_i\|_2^2. \tag{8}$$

### 3.5 Empirical justification

**Domain-invariant learning.** We analyze the impact of domain-invariant learning both qualitatively and quantitatively. Figure 3(a) provides a $t$-SNE visualization (van der Maaten & Hinton, 2008) of domain embeddings from different domains (*i.e.*, Gaussian and Shot noise) before and after applying TestDG, demonstrating that TestDG effectively reduces the gap between different domains. Figure 3(b) shows the Chamfer distance between domain embeddings from different domains before and after applying TestDG, indicating that the distance decreases as intended after the domain-invariant learning.

**Domain embedding.** In Figure 4(a), we empirically verify that domain embeddings capture domain-specific information more prominently than content information. To assess the content information within the domain embeddings, we performed linear probing by training a linear classification layer using the domain embeddings as input while keeping the model frozen. The results show that domain classification using the domain embeddings consistently achieves higher accuracy than class classification, indicating that the domain embeddings predominantly encode domain-specific information with minimal content information, as intended.

**Domain prototype selection.** We qualitatively investigate how well domain prototypes represent the entire set of domain embeddings. Figure 4(b) provides a $t$-SNE visualization of domain embeddings and selected domain prototypes on the Gaussian noise domain, showing that our domain prototypes are diverse and well represent the overall distribution of the domain embeddings.

**Domain prototype update.** Figure 4(c) demonstrates the effectiveness of the prototype updating strategy. By forwarding the corresponding images of each prototype through the updated model, we obtain ideal target prototypes in the updated feature space. The MMD value between these target prototypes and our updated prototypes is lower than those between target prototypes and original prototypes, where MMD values are averaged across all CTTA steps. This result validates that our updating method effectively moves the prototypes toward their targeted representations.

## 4 Experiments

### 4.1 Experimental setting

**Evaluation benchmarks.** Our method was evaluated on both classification and semantic segmentation tasks. For classification, we utilized three benchmarks (Hendrycks & Dietterich, 2019): CIFAR10-C, CIFAR100-C,

Table 1: Classification error rates (%) on CIFAR10-to-CIFAR10-C online CTTA task. Gain (%) represents the percentage of improvement in accuracy compared with the source method.

| Method | gaussian | shot | impulse | defocus | glass | motion | zoom | snow | frost | fog | brightness | contrast | elastic_trans | pixelate | jpeg | Mean↓ | Gain |
|---|---|---|---|---|---|---|---|---|---|---|---|---|---|---|---|---|---|
| Source | 60.1 | 53.2 | 38.3 | 19.9 | 35.5 | 22.6 | 18.6 | 12.1 | 12.7 | 22.8 | 5.3 | 49.7 | 23.6 | 24.7 | 23.1 | 28.2 | / |
| Pseudo | 59.8 | 52.5 | 37.2 | 19.8 | 35.2 | 21.8 | 17.6 | 11.6 | 12.3 | 20.7 | 5.0 | 41.7 | 21.5 | 25.2 | 22.1 | 26.9 | +1.3 |
| TENT | 57.7 | 56.3 | 29.4 | 16.2 | 35.3 | 16.2 | 12.4 | 11.0 | 11.6 | 14.9 | 4.7 | 22.5 | 15.9 | 29.1 | 19.5 | 23.5 | +4.7 |
| CoTTA | 58.7 | 51.3 | 33.0 | 20.1 | 34.8 | 20.0 | 15.2 | 11.1 | 11.3 | 18.5 | 4.0 | 34.7 | 18.8 | 19.0 | 17.9 | 24.6 | +3.6 |
| VDP | 57.5 | 49.5 | 31.7 | 21.3 | 35.1 | 19.6 | 15.1 | 10.8 | 10.3 | 18.1 | 4.0 | 27.5 | 18.4 | 22.5 | 19.9 | 24.1 | +4.1 |
| ViDA | 52.9 | 47.9 | 19.4 | 11.4 | 31.3 | 13.3 | 7.6 | 7.6 | 9.9 | 12.5 | 3.8 | 26.3 | 14.4 | 33.9 | 18.2 | 20.7 | +7.5 |
| Continual-MAE | 30.6 | 18.9 | 11.5 | 10.4 | 22.5 | 13.9 | 9.8 | 6.6 | 6.5 | 8.8 | 4.0 | 8.5 | 12.7 | 9.2 | 14.4 | 12.6 | +15.6 |
| Versatile | 16.3 | 11.1 | 9.6 | 8.4 | 14.6 | 8.6 | 5.5 | 6.3 | 5.7 | 7.1 | 3.3 | 5.4 | 10.9 | 7.7 | 12.8 | 8.9 | +19.3 |
| **TestDG** | 17.4 | 10.9 | 5.4 | 6.3 | 16.3 | 6.9 | 3.9 | 3.7 | 3.8 | 5.9 | 2.2 | 4.7 | 7.8 | 11.2 | 9.4 | **7.7** | **+20.5** |

Table 2: Classification error rates (%) on CIFAR100-to-CIFAR100-C CTTA.

| Method | Mean↓ | Gain |
|---|---|---|
| Source | 35.4 | / |
| Pseudo | 33.2 | +2.2 |
| TENT | 32.1 | +3.3 |
| CoTTA | 34.8 | +0.6 |
| VDP | 32.0 | +3.4 |
| ViDA | 27.3 | +8.1 |
| Continual-MAE | 26.4 | +9.0 |
| Versatile | 24.0 | +11.4 |
| **TestDG** | **23.3** | **+12.1** |

Table 3: Classification error rates (%) on ImageNet-to-ImageNet-C CTTA.

| Method | Mean↓ | Gain |
|---|---|---|
| Source | 55.8 | / |
| Pseudo | 61.2 | -5.4 |
| TENT | 51.0 | +4.8 |
| CoTTA | 54.8 | +1.0 |
| VDP | 50.0 | +5.8 |
| ViDA | 43.4 | +12.4 |
| Continual-MAE | **42.5** | **+13.3** |
| Versatile | 42.7 | +13.1 |
| **TestDG** | **42.5** | **+13.3** |

Table 4: Generalization performance evaluated by error rates (%) on 5 unseen CIFAR10-C domains after applying CTTA over 10 other domains.

| Method | brightness | contrast | elastic_trans | pixelate | jpeg | Mean↓ | Gain |
|---|---|---|---|---|---|---|---|
| Source | 5.3 | 49.7 | 23.6 | 24.7 | 23.1 | 25.3 | / |
| ViDA | 4.1 | 36.4 | 15.6 | 33.5 | 17.8 | 21.5 | +3.8 |
| Continual-MAE | 5.0 | 25.7 | 15.1 | 22.6 | 18.7 | 17.4 | +7.9 |
| **TestDG** | 2.5 | 12.6 | 12.8 | 18.7 | 12.6 | **11.8** | **+13.5** |

Table 5: CTTA results under gradual domain shifts on CIFAR10-to-CIFAR10-C, evaluated by error rates (%).

| | Source | ViDA | Continual-MAE | **TestDG** |
|---|---|---|---|---|
| Mean↓ | 13.7 | 7.7 | 6.5 | **3.6** |
| Gain | / | +6.0 | +7.2 | **+10.1** |

and ImageNet-C, each consisting of 15 corruption types and five severity levels. We reported the results for the most severe level (level 5). For segmentation, we used Cityscapes (Cordts et al., 2016) as the source and ACDC (Sakaridis et al., 2021) as a test dataset, including four unseen test domains: Fog, Night, Rain, and Snow.

**Evaluation scenario.** Following Wang et al. (2022), TestDG was evaluated under an online CTTA setting where the prediction is assessed immediately after data are streamed, and the pre-trained model is adapted on the fly. For the segmentation task, test domains are repeated cyclically for three rounds. In line with recent CTTA studies (Liu et al., 2024b;a; Yang et al., 2024b), *we reported results for all methods using a common ViT backbone.* This follows the standardization introduced by ViDA (Liu et al., 2024b), which established the use of ViTs in CTTA and re-implemented previous CNN-based methods within this ViT-based framework. For additional experiments beyond those reported by previous studies, we evaluated methods with publicly available code.

**Implementation details.** We utilized ViT-base (Dosovitskiy et al., 2021) for classification tasks and Segformer-B5 (Xie et al., 2021) for the segmentation task as the backbone. Input images were resized to $384 \times 384$ for CIFAR10-C and CIFAR100-C, $224 \times 224$ for ImageNet-C, and to $960 \times 540$ for ACDC. The size of the queue storing the domain embeddings and domain prototypes was set to 256 and 40, respectively. The domain information extractor $\psi_{\text{domain}}$ and domain discriminator $D$ were both implemented as MLP whose activation functions are ReLU (Agarap, 2018) and GELU (Hendrycks & Gimpel, 2016), respectively. The domain amplifier was implemented as an adapter that includes an up-projection layer and a down-projection layer, with an intermediate dimension of 128.

Table 6: Segmentation mIoU results on Cityscape-to-ACDC online CTTA task.

| Time | $t$ ———————————————————→ | | | | | | | | | | | | | | | | Mean↑ |
|---|---|---|---|---|---|---|---|---|---|---|---|---|---|---|---|---|---|
| Round | 1 | | | | | 2 | | | | | 3 | | | | | | |
| Method | Fog | Night | Rain | Snow | Mean↑ | Fog | Night | Rain | Snow | Mean↑ | Fog | Night | Rain | Snow | Mean↑ | |
| Source | 69.1 | 40.3 | 59.7 | 57.8 | 56.7 | 69.1 | 40.3 | 59.7 | 57.8 | 56.7 | 69.1 | 40.3 | 59.7 | 57.8 | 56.7 | 56.7 |
| TENT | 69.0 | 40.2 | 60.1 | 57.3 | 56.7 | 68.3 | 39.0 | 60.1 | 56.3 | 55.9 | 67.5 | 37.8 | 59.6 | 55.0 | 55.0 | 55.7 |
| CoTTA | 70.9 | 41.2 | 62.4 | 59.7 | 58.6 | 70.9 | 41.1 | 62.6 | 59.7 | 58.6 | 70.9 | 41.0 | 62.7 | 59.7 | 58.6 | 58.6 |
| DePT | 71.0 | 40.8 | 58.2 | 56.8 | 56.5 | 68.2 | 40.0 | 55.4 | 53.7 | 54.3 | 66.4 | 38.0 | 47.3 | 47.2 | 49.7 | 53.4 |
| ECoTTA | 68.5 | 35.8 | 62.1 | 57.4 | 56.0 | 68.3 | 35.5 | 62.3 | 57.4 | 55.9 | 68.1 | 35.3 | 62.3 | 57.3 | 55.8 | 55.8 |
| VDP | 70.5 | 41.1 | 62.1 | 59.5 | 58.3 | 70.4 | 41.1 | 62.2 | 59.4 | 58.2 | 70.4 | 41.0 | 62.2 | 59.4 | 58.2 | 58.2 |
| ViDA | 71.6 | 43.2 | 66.0 | 63.4 | 61.1 | 73.2 | 44.5 | 67.0 | 63.9 | 62.2 | 73.2 | 44.6 | 67.2 | 64.2 | 62.3 | 61.9 |
| BECoTTA | 72.3 | 42.0 | 63.5 | 60.1 | 59.5 | 72.4 | 41.9 | 63.5 | 60.2 | 59.5 | 72.3 | 41.9 | 63.6 | 60.2 | 59.5 | 59.5 |
| SVDP | 72.1 | 44.0 | 65.2 | 63.0 | 61.1 | 72.2 | 44.5 | 65.9 | 63.5 | 61.5 | 72.1 | 44.2 | 65.6 | 63.6 | 61.4 | 61.3 |
| Continual-MAE | 71.9 | 44.6 | 67.4 | 63.2 | 61.8 | 71.7 | 44.9 | 66.5 | 63.1 | 61.6 | 72.3 | 45.4 | 67.1 | 63.1 | 62.0 | 61.8 |
| Zhu *et al.* | 71.2 | 42.3 | 64.9 | 62.0 | 60.1 | 72.6 | 43.2 | 66.3 | 63.2 | 61.3 | 72.8 | 43.8 | 66.5 | 63.2 | 61.6 | 61.0 |
| **TestDG** | 71.9 | 45.5 | 66.4 | 63.9 | **62.0**$_{\pm 0.1}$ | 73.2 | 45.8 | 67.4 | 64.1 | **62.6**$_{\pm 0.1}$ | 72.8 | 44.7 | 67.9 | 63.7 | **62.3**$_{\pm 0.1}$ | **62.3**$_{\pm 0.1}$ |

## 4.2 Evaluation on classification tasks

To evaluate the performance of TestDG, we conducted extensive experiments on three conventional benchmarks: CIFAR10-C, CIFAR100-C, and ImageNet-C.

**Performance on corruption benchmarks.** Table 1 and Table 2 present the results on CIFAR10-C and CIFAR100-C. TestDG significantly outperformed all other methods across various corruption types. Specifically, it achieved the lowest mean error rate of 7.7% on CIFAR10-C and 23.3% on CIFAR100-C, showing improvements of 20.5%p and 12.1%p over the source model, respectively. As shown in Table 3, TestDG also achieved the best performance on the ImageNet-C dataset, where performance differences are naturally smaller due to the large number of classes.

**Generalization on unseen domains.** We evaluated the generalization ability to unseen domains on CIFAR10-C using the leave-one-domain-out rule (Zhou et al., 2021; Liu et al., 2024b), where CTTA methods are applied to 10 domains and tested on 5 unseen domains. Table 4 shows that TestDG achieved the lowest mean error rate, highlighting the benefit of domain-invariant learning and its generalization ability even on unseen domains.

**Gradually changing results.** Table 5 shows results of continual test-time adaptation on a CIFAR10-C gradually changing setup (Wang et al., 2022), where the corruption types changed gradually over time. In this more realistic scenario where domain boundaries are blurred, TestDG still demonstrated superior performance, notably outperforming existing methods with a mean error rate of 3.6%. This result highlights the effectiveness of domain-invariant learning even under gradual domain transitions.

## 4.3 Evaluation on segmentation tasks

Table 6 presents the segmentation results from adapting Cityscapes to ACDC, a real-world adverse condition dataset consisting of four challenging domains: Fog, Night, Rain, and Snow. Across three rounds, TestDG consistently achieved the highest mIoU among existing CTTA methods. In the first round, our method reached an average mIoU of 62.0%, showing a 5.3%p gain over the source model. In subsequent rounds, TestDG performed strongly on newly encountered domains while preserving robustness on previously encountered domains, effectively handling continually changing environments. Notably, TestDG maintained consistent improvements across all four domain types, indicating that domain-invariant learning generalizes well beyond classification to dense prediction tasks. These results suggest that TestDG successfully mitigates domain shifts in semantic segmentation tasks, maintaining robustness across diverse and real-world adverse conditions.

Table 7: Ablation study on each component of TestDG.

| $\psi_{\text{amplifier}}$ | | | | | Mean↓ |
|---|---|---|---|---|---|
| | | | | | 21.9 |
| | ✓ | | | | **7.7** |

| $\mathcal{L}_{\text{self}}$ | $\mathcal{L}_{\text{inv}}$ | $\mathcal{L}_{\text{dis}}$ | $J(\mathcal{P}_{\text{pre}})$ | $\mathcal{L}_{\text{update}}$ | Mean↓ |
|---|---|---|---|---|---|
| | | | | | 28.2 |
| ✓ | | | | | 18.0 |
| ✓ | ✓ | | | | 10.5 |
| ✓ | ✓ | ✓ | | | 8.3 |
| ✓ | ✓ | ✓ | ✓ | | 8.0 |
| ✓ | ✓ | ✓ | ✓ | ✓ | **7.7** |

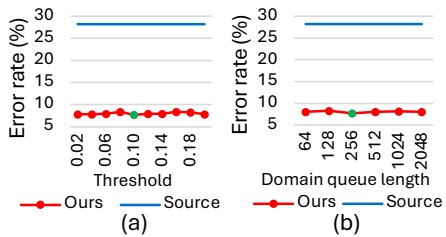

Figure 5: Sensitivity to hyperparameters. (a) Threshold for domain change detection. (b) Queue length.

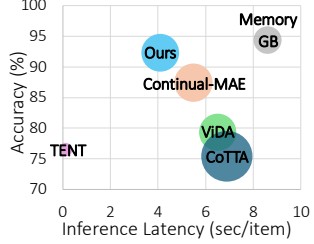

Figure 6: Comparison of computational cost on CIFAR10-C.

## 4.4 In-depth analysis

**Impact of domain amplifier.** We investigated the impact of the domain amplifier by evaluating TestDG with and without it. As shown in Table 7, removing the domain amplifier significantly degrades performance, with the error rate increasing from 7.7% to 21.9%. This demonstrates that the domain amplifier is essential for effective test-time adaptation, as it extracts domain-specific information from each layer of the encoder, thereby enabling the encoder to focus on learning domain-invariant features. The separation of roles between domain-specific extraction (via the amplifier) and domain-invariant learning (via the encoder) is crucial for robust continual test-time adaptation.

**Ablation study of each loss.** We conducted an ablation study to evaluate the contributions of each loss component in our method. The results, presented in Table 7, demonstrate that each loss contributes to the overall performance improvement. Notably, the domain-invariant learning loss $\mathcal{L}_{\text{inv}}$ was crucial for achieving high performance by guiding the model to learn domain-invariant features, which are essential for effective adaptation across different domains. Additionally, the domain discrimination loss $\mathcal{L}_{\text{dis}}$, which facilitated the learning of domain embeddings, was also crucial for TestDG. Furthermore, both the score function for domain prototype selection $J(\mathcal{P}_{\text{pre}})$ and the prototype updating loss $\mathcal{L}_{\text{update}}$ contributed to the performance of TestDG.

**Sensitivity to the threshold for domain change detection.** To investigate the effect of the threshold used for domain change detection, we conducted experiments varying the threshold values. Figure 5(a) summarizes the results for thresholds ranging from 0.02 to 0.20. The performance remained stable across this range of thresholds, consistently outperforming the source model. These results suggest that TestDG is insensitive to threshold settings, which is practically important since the optimal threshold may vary across different deployment scenarios.

**Sensitivity to the queue length $|\mathcal{F}_{\text{pre}}|$.** We evaluated the impact of $|\mathcal{F}_{\text{pre}}|$, which specifies the length of the queue that stores previous domain embeddings. It is designed to retain a sufficient number of embeddings to capture the distribution of the previous domain while minimizing the inclusion of excessively outdated information. Figure 5(b) presents the error rates on CIFAR10-C for six different queue lengths ranging from 64 to 4,096. The results indicate that the error rates remain consistent across all tested values of $|\mathcal{F}_{\text{pre}}|$, demonstrating that TestDG is insensitive to the length of the queue and can maintain strong performance even with a compact memory footprint.

**Comparison of computational cost.** We evaluated the computational cost of TestDG in terms of inference latency and memory consumption alongside prediction accuracy. As shown in Figure 6, while TENT (Wang et al., 2021a) exhibited the highest computational efficiency, it had the lowest prediction accuracy among the evaluated methods. In contrast, TestDG achieved reduced inference latency and comparable memory consumption compared to other CTTA methods (excluding TENT) while significantly outperforming them in prediction accuracy. This favorable trade-off between computational cost and performance makes TestDG practical for real-world deployment where both efficiency and robustness are required.

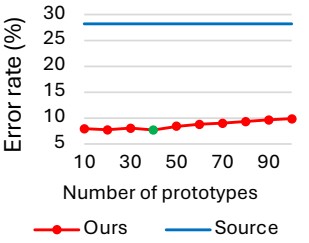

Figure 7: Ablation on the number of domain prototypes.

Table 8: Distance metrics for prototype update.

| Metric | MMD | Hausdorff | Chamfer |
|--------|-----|-----------|---------|
| Error  | 8.2 | 8.1       | **7.7** |

Table 9: Extended ablation on thresholds.

| Length | 8192 | 16384 | 32768 |
|--------|------|-------|-------|
| Error  | 8.1  | 8.0   | 8.1   |

(a) Queue length

| Conf  | 0.4 | 0.6 | 0.8 |
|-------|-----|-----|-----|
| Error | 8.1 | 8.3 | 8.1 |

(b) Confidence threshold

**Number of domain prototypes.** We conduct an ablation study on the number of domain prototypes to analyze its impact on TestDG's performance. As shown in Figure 7, we vary the number of domain prototypes from 10 to 100 while keeping other hyperparameters fixed. Having too few prototypes leads to degraded performance as they cannot sufficiently capture the domain characteristics. However, using too many prototypes not only increases computational overhead but also risks including outliers that may degrade the quality of domain-invariant learning. We find that TestDG is insensitive to the number of domain prototypes, maintaining strong performance across a wide range, which suggests that TestDG can achieve robust performance without requiring careful tuning of this hyperparameter.

**Distance metrics for prototype update.** We investigate the effectiveness of different distance metrics (*i.e.*, MMD, Hausdorff, and Chamfer) for updating the prototypes in the domain embedding space. As shown in Table 8, Chamfer distance results in the lowest error among the three metrics. This is because Chamfer distance is sensitive to individual point shifts, which more effectively ensures that each prototype remains consistent in the domain embedding space before and after model updates. In contrast, MMD and Hausdorff distance are less sensitive to individual point movements, leading to slightly worse performance.

**Extended ablation on thresholds.** We expand the ablation studies to include a wider range of queue lengths and domain change detection thresholds beyond those reported in Figure 5. Table 9 shows that TestDG remains robust to a very wide range of hyperparameter settings, including queue lengths up to 32,768 and confidence thresholds up to 0.8, demonstrating the practical reliability of our method.

**Comparison of learnable parameters.** We compare the number of learnable parameters between TestDG and other CTTA methods in Table 10. TestDG requires only 7.0M additional parameters compared to TENT and CoTTA, which is a modest increase considering the significant performance improvement it achieves. Despite having more parameters than some baselines, TestDG achieves substantially better performance, demonstrating that the additional parameters are used effectively to capture domain-specific features and enhance adaptation.

**Statistical significance analysis.** To rigorously evaluate the effectiveness of our method, we perform statistical significance testing by conducting paired t-tests between TestDG and existing methods on the ACDC dataset. Table 11 shows that all p-values fall well below the standard significance level of 0.05, with values ranging from $10^{-8}$ to $10^{-3}$. These results strongly support that TestDG achieves meaningful performance gains over existing methods in continual test-time adaptation, rather than improvements due to random variation.

**Analysis on domain amplifier.** We further examine how the domain amplifier influences attention heatmaps under domain-invariant learning. Figure 8 compares attention heatmaps of models with and without the domain amplifier across different domains. Enabling the domain amplifier consistently yields more focused and reliable attention maps, even under challenging environmental shifts. This improvement is especially pronounced in regions where domain-specific variations (*e.g.*, lighting and texture) are prominent, suggesting that the amplifier fosters stronger feature alignment and focuses the model's attention on consistently relevant regions. Figure 9 illustrates how the domain amplifier is integrated into each layer of the encoder. Following

Table 10: Error rates and learnable parameters.

| Method | Error (%) | Params |
|---|---|---|
| TENT | 23.5 | 0.038M |
| CoTTA | 24.6 | 86.1M |
| ViDA | 20.7 | 93.2M |
| Continual-MAE | 12.6 | 86.1M |
| **TestDG** | **7.7** | 93.1M |

Table 11: Statistical significance (p-values) on ACDC.

| Comparison | p-value | Comparison | p-value |
|---|---|---|---|
| Source vs. Ours | $6.77\times10^{-8}$ | ViDA vs. Ours | $2.30\times10^{-3}$ |
| TENT vs. Ours | $3.51\times10^{-8}$ | BECoTTA vs. Ours | $1.08\times10^{-6}$ |
| CoTTA vs. Ours | $3.55\times10^{-7}$ | SVDP vs. Ours | $6.52\times10^{-5}$ |
| DePT vs. Ours | $1.06\times10^{-8}$ | Continual-MAE vs. Ours | $9.80\times10^{-4}$ |
| ECoTTA vs. Ours | $3.73\times10^{-8}$ | Zhu *et al.* vs. Ours | $2.30\times10^{-5}$ |
| VDP vs. Ours | $2.60\times10^{-7}$ | | |

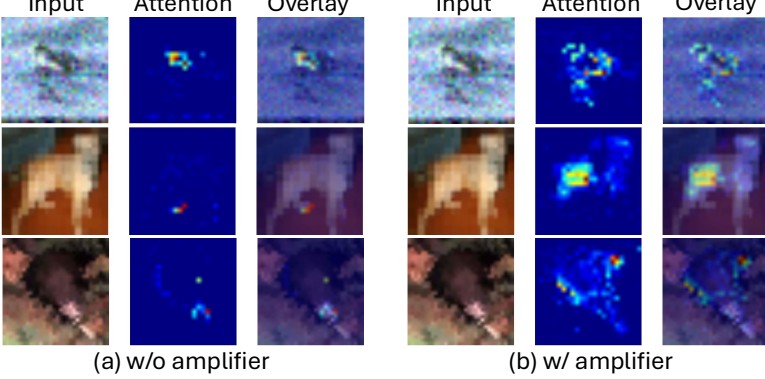

Figure 8: Domain amplifier analysis. (a) Without amplifier. (b) With amplifier.

Figure 9: Domain amplifier structure.

the conventional adapter structure for parameter-efficient fine-tuning (Chen et al., 2022; Houlsby et al., 2019; Pfeiffer et al., 2020), the domain amplifier uses a bottleneck structure (down-projection, then up-projection) to capture domain-specific information while keeping parameter overhead minimal. Attaching a domain amplifier to each encoder layer allows the model to incorporate domain cues at multiple semantic levels, facilitating robust adaptation in non-stationary test settings.

# 5   Conclusion

We have introduced TestDG, a novel online test-time domain generalization framework for CTTA. Unlike existing CTTA methods that focus only on adaptation to the current test domain and thus often lack generalization to arbitrary future test domains, TestDG aims to learn domain-invariant features during testing through a carefully designed model architecture and test-time training strategy. To achieve this, we proposed a domain information extractor and domain amplifier that separate domain-specific information from encoder features, along with a domain prototype selection mechanism based on submodular optimization and a Chamfer distance-based prototype update strategy. Through extensive experiments on four public CTTA benchmarks, including both classification (CIFAR10-C, CIFAR100-C, ImageNet-C) and semantic segmentation (Cityscapes-to-ACDC) tasks, we have validated that TestDG improves generalization to arbitrary unseen domains, ensures robust adaptation in dynamically changing test environments, and consequently achieves state-of-the-art performance. Furthermore, comprehensive ablation studies and statistical analyses confirm the contribution of each component and the significance of the improvements.

**Limitations and future work.** TestDG is less robust to initial corruptions due to limited exposure to diverse domain shifts at the early stages of adaptation, and we believe early adaptation strategies could mitigate this issue. Also, TestDG currently uses only the most recent previous domain for domain-invariant learning. Incorporating information from additional previously encountered domains with minimal memory overhead could further improve its domain generalization capability and is a promising direction for future work.

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
