# OpenReview forum: "TestDG: Test-time Domain Generalization for Continual Test-time Adaptation"
_TMLR — Under review for TMLR_

### Review · Reviewer_vUTx · 2026-05-10

**Summary Of Contributions:**

The paper introduces TestDG, a framework designed for Continual Test-Time Adaptation (CTTA). While existing CTTA methods primarily focus on adapting models to the immediate test environment, TestDG aims to facilitate on-the-fly domain generalization to prepare the model for future, unseen domains. The framework attempts to extract features that remain invariant across both current and historical domains. To support this, the authors propose an adapted model architecture, alongside a specific data structure and optimization algorithm intended to manage historical domain information. The proposed method is evaluated across four public CTTA benchmarks, achieving competitive empirical results.

**Audience:**

Yes

**Audience Explanation:**

The paper targets Continual Test-Time Adaptation (CTTA), which is a highly relevant and actively researched topic within the TMLR and broader machine learning communities.

**Broader Impact Concerns:**

No specific ethical concerns or broader impact issues were identified in this work. The research focuses on algorithmic robustness and generalization in machine learning models, which does not inherently pose direct negative societal impacts.

**Claims And Evidence:**

Yes

**Claims Explanation:**

While the empirical benchmark results indicate effectiveness, several core claims regarding the methodology lack convincing evidence or clear differentiation from existing literature:

W1.   The authors claim a "new model architecture and a test-time adaptation strategy dedicated to learning domain-invariant features." However, learning invariant features across historical domains is fundamentally identical to existing Domain Generalization (DG) and Continual Learning (CL) paradigms. The submission lacks clear theoretical evidence proving how this strategy differs structurally from merely combining existing CL alignment techniques at test time.

W2.   The abstract heavily relies on the claim of a "new data structure... for effectively managing information from previous test domains." Functionally, this mechanism strongly resembles standard episodic memory buffers or rehearsal queues. Without an explicit algorithmic ablation comparing this "new" structure to standard memory buffers, the claim of novelty here is unsupported.

W3.  The framework claims to optimize and extract invariant features "on the fly during testing." Enforcing multi-domain invariance typically incurs high computational overhead. The submission currently lacks a rigorous computational complexity analysis (e.g., FLOPs, inference latency, memory overhead) to convincingly support the claim that this algorithm is feasible for real-time test adaptation.

W4. The claim of "superior generalization to unseen test domains" is insufficiently proven. Because the framework relies on "effectively managing information from previous test domains," the extracted invariant features are inherently conditioned on the sequence of encountered environments. The evaluation must demonstrate robustness against sequence-ordering bias (e.g., via permutation experiments) to fully support the claim of generalized robustness.

**Requested Changes:**

1. Clarify the differences between the proposed TestDG framework and existing Domain Generalization (DG) or Continual Learning (CL) paradigms. Furthermore, provide a rigorous algorithmic definition of the "new data structure" to clearly distinguish it functionally from conventional episodic memory buffers.

2. Supplement the empirical section with deeper analyses to substantiate the core claims. This should include targeted ablation studies isolating the performance contribution of the proposed data structure, sequence permutation experiments to verify generalization robustness against domain-ordering bias, and a computational overhead analysis (e.g., FLOPs, inference latency) to prove the real-world feasibility of the "on-the-fly" optimization.

---

> ### Author Response · Authors · 2026-07-06
>
> We appreciate your insightful feedback and constructive suggestions that help improve our paper substantially. We will address all the comments and include the additional experiments in the revision. Please find our responses below. For experiments beyond those reported in previous studies, we evaluated the compared methods using their publicly available code, and unless noted otherwise the ablations were conducted on CIFAR10-C.
>
> ___
> ### **1. Difference from existing DG and CL paradigms**
>
> Thank you for the comment. Both DG and CL assume access to more than one domain at once. DG trains on multiple labeled source domains, and CL assumes access to past tasks through stored data or known task boundaries. CTTA breaks this assumption, since only the current unlabeled domain is available at each step and every previous domain has passed and thus is inaccessible. Enabling alignment under this constraint is our contribution, and TestDG achieves it through three mechanisms. Instead of the raw data of the previous domain, it stores a small set of representative prototypes (Sec. 3.2). Instead of the raw features, it separates and aligns only the domain embeddings, preserving the semantics (Eq. 5-6, Fig. 4(a)). And it continuously updates the prototypes as the model drifts (Eq. 7). These mechanisms make it possible to reduce the divergence between domains online, which in turn encourages domain-invariant features, following the principle that smaller inter-domain divergence improves generalization [1, 2].
>
> [1] Ben-David et al. A theory of learning from different domains. (Machine Learning, 2010)
>
> [2] Albuquerque et al. Generalizing to unseen domains via distribution matching. (arXiv, 2019)
> ___
>
> ### **2. Distinction from standard memory buffers**
>
> Thank you for the comment. We will tone down the novelty claim of our data structure and clarify how it differs from a standard rehearsal buffer. It differs in three ways. It stores low-dimensional domain embeddings rather than raw samples, keeps only the most recent domain so the memory stays constant regardless of the stream length, and selects representative prototypes rather than sampling at random, by FIFO, or by reservoir sampling. The procedure is summarized below.
>
> **Algorithm.**
> 1. Maintain a set $P_{pre}$ of $n = 40$ domain prototypes for the previous domain.
> 2. Every test batch, update $P_{pre}$ via the Chamfer objective (Eq. 7) so it tracks the drifting feature space.
> 3. At a domain change, reselect the $n$ prototypes that best represent the previous domain by the submodular score $J(P_{pre})$ (Eq. 3), and replace $P_{pre}$.
>
> Following your suggestion, we replaced our data structure with standard buffers under the same memory budget. As shown below, our selection best approximates the previous-domain distribution, measured by the squared MMD between the selected prototypes and the previous-domain embeddings, and yields the lowest CTTA error, so the gain comes from the data structure rather than a plain buffer. We will include these results in the revision.
>
> | Selection strategy | Mean error | MMD² to previous domain |
> |:---:|:---:|:---:|
> | FIFO | 8.8 | 0.0256 |
> | Random | 8.6 | 0.0248 |
> | Reservoir | 8.4 | 0.0251 |
> | Ours | **7.7** | **0.0216** |
>
> ___
> ### **3. Computational complexity**
>
> Since domain-invariant learning is performed only between the previous and current domains, the memory and per-batch cost do not grow as more domains are encountered. We conducted an in-depth analysis of computational complexity on CIFAR10-C with a batch size of 40 on a single NVIDIA RTX A6000; the results are reported in the table below. The table shows that TestDG achieves the fewest FLOPs and the lowest latency per step at comparable memory, since it uses a single augmented view whereas CoTTA and ViDA average the teacher over many views. This suggests that TestDG is more efficient than existing CTTA methods despite performing domain-invariant learning at test time. We will include this analysis in the revision.
>
> | Method | Forwards/step | TFLOPs/step | Latency (sec/item) | GPU memory |
> |:---:|:---:|:---:|:---:|:---:|
> | CoTTA | 35 | 82.20 | 6.8 | 47.3 GB |
> | ViDA | 12 | 31.10 | 6.5 | 33.7 GB |
> | Continual-MAE | 12 | 31.10 | 5.5 | 36.5 GB |
> | TestDG | **4** | **13.33** | **4.2** | 34.4 GB |
>
> ___
> ### **4. Robustness to domain ordering**
>
> Since TestDG learns domain-invariant features, it is insensitive to the domain order. Following your suggestion, we ran 4 different corruption orders on CIFAR10-C. As shown below, the standard deviation is clearly below our 1.2%p improvement over the previous state of the art, showing that the gain is stable. We will report these results in the revision.
> | Setting | Mean error |
> |:---:|:---:|
> | 4 corruption orders | 7.87 ± 0.40 |

---

### Review · Reviewer_h7WC · 2026-06-13

**Summary Of Contributions:**

The authors propose a novel test time domain generalization framework where the goal is to learn features invariant to current and future domains on the fly during test time.

### Strengths:
- The motivation for this paper is well justified, as they focus on both adaptation and generalization to future domains.
- The need and the role of proposed components, like domain invariant learning, use of domain prototypes and test objectives are well justified.
- The experimental analysis, including classification and segmentation benchmarks, continual and gradual TTA setting, ablation positively support the method's effectiveness.

### Weaknesses:
- The paper's central innovation is generalization to future domains, yet Table 4 is the only evaluation done in this context (CIFAR10-C with 5 held-out corruptions). No equivalent experiment exists for CIFAR100-C, ImageNet-C, or ACDC. Given that this is the main motivation, it needs more extensive experimental analysis in this regard.
- The prototypes only from the previous domain are stored. Why not store prototypes from evan earlier domains to promote generalization which is the primary motivation? This design choice is not well supported and should be explored more rigorously.
- According to Figure 6, the latency of TestDG is lower than CoTTA, ViDA. Why? TestDG includes several steps including two alternating optimization steps, prototype updation, which is more complex that CoTTA. Can this be justified in terms of Flops/ forward and backward passes, batch size etc.

**Audience:**

Yes

**Audience Explanation:**

The paper addresses a meaningful problem about learning domain-invariant features during test time to generalize to future unseen domains. This work is at an intersection of test time adaptation, domain generalization, which is of interest in TMLR's audience.

**Claims And Evidence:**

No

**Claims Explanation:**

- A prior work T3A[1] formulates test domain generalization and the claim on novel test time domain generalization framework has to be toned down . It is important to discuss this and include as a baseline as it is highly relevant in this context.
- Although the paper frames test time domain generalization to future unseen domains as a central motivation, most benchmarks evaluate adaptation to the current domain. Only Table 4 evaluates test domain generalization for one dataset (CIFAR-10C) and compared against only two baselines. The narrative is not coherent with the experiments done.

[1] Yusuke Iwasawa and Yutaka Matsuo. Test-time classifier adjustment module for model-agnostic domain generalization. NeurIPS, 2021.

**Requested Changes:**

- Expand the domain generalization evaluation (currently only Table 4) to include CIFAR100-C, ImageNet-C, and ACDC, and compare against a broader set of baselines including T3A [1]. This is critical given that future domain generalization is the paper's primary motivation.
- Discuss T3A [1] in the related work and include it as a baseline in all generalization experiments. The claim of a novel test-time domain generalization framework must be reframed in the context of T3A.
- Justify and empirically analyze the design choice of storing prototypes from only the most recent previous domain. An ablation comparing single vs. multiple previous domain prototypes should be provided, as retaining earlier domain prototypes may better serve the generalization objective.
- Provide a rigorous computational cost analysis including FLOPs, number of forward and backward passes, and batch size details to support the latency claims in Figure 6. The current scatter plot is insufficient.

---

> ### Author Response · Authors · 2026-07-06
>
> We appreciate your insightful feedback and constructive suggestions that help improve our paper substantially. We will address all the comments and include the additional experiments in the revision. Please find our responses below. For experiments beyond those reported in previous studies, we evaluated the compared methods using their publicly available code, and unless noted otherwise the ablations were conducted on CIFAR10-C.
>
>
> ___
> ### **1. Additional domain generalization experiments**
>
>
> Thank you for the valuable suggestion. We extended the generalization evaluation of Table 4 to CIFAR100-C and ImageNet-C with an extra baseline, T3A, and additionally evaluated the methods on the real-world ACDC benchmark under the leave-one-domain-out protocol. T3A is included only on the classification benchmarks (CIFAR10-C, CIFAR100-C, ImageNet-C) since it is not applicable to segmentation (ACDC). As shown below, TestDG achieved the best generalization on every benchmark, including the real-world domains of ACDC. We will include these results in the revision.
>
>
> | | Source | T3A | ViDA | TestDG |
> |:---:|:---:|:---:|:---:|:---:|
> | CIFAR10-C (Table 4) | 25.3 | 13.6 | 21.5 | **11.8** |
> | CIFAR100-C | 45.2 | 34.8 | 49.5 | **33.1** |
> | ImageNet-C | 49.9 | 42.2 | 40.6 | **39.9** |
>
> | | Source | CoTTA | ViDA | TestDG |
> |:---:|:---:|:---:|:---:|:---:|
> | ACDC | 58.3 | 58.5 | 58.4 | **59.0** |
>
>
>
> ___
> ### **2. Comparison with T3A**
>
>
> Thank you for the comment. We will cite and discuss T3A in the related work, add it as a baseline in all generalization experiments (Response 1), and tone down the claim by framing TestDG as test-time domain generalization in the continual setting. Here, we discuss the key difference between TestDG and T3A: T3A adapts to the current test domain without considering previous domains, while TestDG instead uses information from previous domains to learn domain-invariant features that generalize to future unseen domains. As reported in the first response, this leads to stronger generalization of TestDG, and consequently allows it to surpass  T3A on the generalization benchmarks. We will add this discussion in the revision.
> ___
> ### **3. Using prototypes from multiple previous domains**
>
>
> Thank you for the great comment. As shown in the table below, using prototypes of more previous domains worsens both adaptation and generalization. During CTTA, the feature space is continually updated, so prototypes from earlier domains grow outdated and no longer represent their domains, and aligning to such stale references harms generalization. Retaining only the most recent domain thus performs best while keeping the memory constant regardless of the stream length. We will include this ablation in the revision.
>
>
> | Number of previous domains | CTTA | DG |
> |:---:|:---:|:---:|
> | 1 (ours) | **7.7** | **11.8** |
> | 3 | 8.3 | 12.7 |
> | 5 | 8.2 | 13.1 |
>
>
>
>
> ___
> ### **4. Latency compared to CoTTA and ViDA**
>
> The lower latency of our method over CoTTA and ViDA does not come from having fewer optimization steps but from the lower number of forward passes. As shown below, CoTTA and ViDA average the teacher over many augmented views, which dominates their cost, whereas TestDG uses a single augmented view. All methods run one backward pass at the same batch size of 40, so the difference is driven by the forward count. TestDG thus attains the lowest latency and FLOPs. We will complement Figure 6 with a detailed table in the revision.
>
> | Method | Forwards/step | Backwards/step | TFLOPs/step | Latency (sec/item) |
> |:---:|:---:|:---:|:---:|:---:|
> | CoTTA | 35 | 1 | 82.20 | 6.8 |
> | ViDA | 12 | 1 | 31.10 | 6.5 |
> | TestDG | **4** | 1 | **13.33** |**4.2**|

---

### Review · Reviewer_aCK8 · 2026-06-23

**Summary Of Contributions:**

The current manuscript introduces *TestDG* -- a continual test-time adaptation framework-- that explicitly targets test-time domain generalization rather than only adaptation to the current domain. The main idea is to learn domain-invariant representations online during testing by extracting domain-specific embeddings from current test data, storing representative prototypes from previous domains, and aligning current-domain embeddings with these prototypes. TestDG introduces a *domain information extractor, a domain amplifier, a prototype-selection mechanism based on submodular optimization/MMD, and a prototype-update strategy based on Chamfer distance. The paper evaluates the method on some benchmarks, such as CIFAR10-C, CIFAR100-C, ImageNet-C, and Cityscapes-to-ACDC, and reports competitive performance.

**Additional Comments:**

I would consider the paper promising but needing clarification. The empirical results are good, and the idea is somehow interesting. However, before acceptance, I would require clarification of 1) the loss formulation, 2) the prototype matching/update mechanism, 3) fairer experimental controls, and 4) stronger statistical evidence. The most critical technical issue is the apparent sign problem in the domain-discrimination loss and the ambiguity of the invariant alignment loss.

**Audience:**

Yes

**Audience Explanation:**

Yes. The paper addresses a topic that is likely to be of interest to at least part of the TMLR audience, especially researchers working on test-time adaptation, domain generalization, robustness under distribution shift, continual learning, and reliable deployment of vision models. The problem setting is practically relevant: deployed models often face non-stationary and previously unseen test environments, and improving robustness in such scenarios is an important research direction. The paper’s main finding—that CTTA can be improved by explicitly encouraging domain-invariant representations during test-time adaptation—is interesting and potentially useful. The empirical results on both classification and segmentation benchmarks suggest that this direction may lead to improved robustness compared with existing CTTA methods.

**Claims And Evidence:**

No

**Claims Explanation:**

The main empirical claims are somewhat supported by the evidence provided, but some of the stronger conceptual claims would benefit from further clarification and experimentation.

The paper includes experimental evidence on several standard CTTA benchmarks ( CIFAR10-C, CIFAR100-C, ImageNet-C, and Cityscapes-to-ACDC). The reported results show good performance compared with several relevant baselines, and the ablation studies support the usefulness of the main components, including the domain-invariant loss, domain-discrimination loss, prototype selection, prototype update, and domain amplifier. The additional analysis on threshold sensitivity, queue length, number of prototypes, computational cost, and statistical significance on ACDC further strengthens the empirical case.

But, I am not fully convinced that all claims are supported with sufficient clarity:
- First, the claim of improved “domain generalization” is mainly demonstrated through held-out **corruption types**, which is useful but still *limited evidence for generalization to arbitrary future test domains*.
- second, some methodological details are unclear, especially the exact formulation of the domain-discrimination loss and how current domain embeddings are paired or matched with previous prototypes in the invariant loss. These details are important for evaluating whether the proposed mechanism truly learns domain-invariant representations rather than simply benefiting from additional adaptation capacity.
- Thid, most main results are reported without standard deviations over *multiple seeds* and *multiple domain orders*, except for limited significance analysis on ACDC. This makes it difficult to assess the robustness of the reported improvements.

**Requested Changes:**

Can you please carefully read these questions and provide answers in your text:
 1) How exactly is the domain discrimination loss implemented? (Is Eq. 5 a typo? What is the exact binary cross-entropy objective used in code?)
2)  How are $f_i$ and $p_i$ paired in the invariant loss? (are prototypes matched randomly, by nearest neighbors, by index, or by a set-distance objective?)
3) Does the improvement come from domain generalization or simply from adding a powerful adapter? have you compared against an adapter-only baseline with the same number of trainable parameters?
4) How robust is the method to wrong domain-change detection ?Can you compare confidence-based detection against oracle boundaries and against no detection?
5) how robust is the method under label shift or class imbalance? (Since it uses pseudo-labeling and confidence, this could be a failure mode).
6) Would storing prototypes from multiple previous domains improve generalization or reduce forgetting?
7) How many random seeds and domain orders were used? Are the reported gains stable across different corruption sequences?
8) Why is the ImageNet-C result claimed as best when it appears tied with Continual-MAE?
9) What are the exact computational settings?
10) Can you provide more details on hardware, batch size, update steps, and memory measurements used for the latency plot?
11) Can you provide class-conditional alignment analysis? (This would help show that domain-invariant learning does not erase semantic class information.)

---

> ### Author Response · Authors · 2026-07-06
>
> We appreciate your insightful feedback and constructive suggestions that help improve our paper substantially. We will address all the comments and include the additional experiments in the revision. Please find our responses below. For experiments beyond those reported in previous studies, we evaluated the compared methods using their publicly available code, and unless noted otherwise the ablations were conducted on CIFAR10-C.
> Responses 1 to 11 follow your requested changes in order. Response 12 addresses your concern about generalization beyond held-out corruptions.
> ___
> ### **1. Typo in Eq. 5**
>
> The sign of the second term in Eq. 5 is wrong, and we sincerely apologize for this typo. In our implementation, the discriminator $D$ is trained to assign 1 to the current-domain embeddings $F_{cur}$ and 0 to the previous-domain prototypes $P_{pre}$, which is the standard binary cross-entropy. Hence, the corrected loss should be
>
> $$L_{dis}(F_{cur}, P_{pre}) = -\\frac{1}{n} \\left[ \\sum_{f_i \\in F_{cur}} \\log D(f_i) + \\sum_{p_i \\in P_{pre}} \\log(1 - D(p_i)) \\right].$$
>
> We will correct the equation in the revision.
> ___
> ### **2. Pairing of $f_i$ and $p_i$ in $L_{inv}$**
>
> In $L_{inv}$ (Eq. 6), $f_i$ and $p_i$ are paired at random. A specific correspondence between $F_{cur}$ and $P_{pre}$ is not required since they are designed to represent domain properties only, not the class information (Fig. 4(a)). Hence, random pairing reduces the domain gap in practice. We will add this detail around Eq. 6 in the revision.
> ___
> ### **3. Comparison with an adapter-only baseline**
>
> Thank you for the valuable suggestion. We compare an adapter-only baseline with the same number of trainable parameters against TestDG. On CIFAR10-C this is reported in Table 7, and the same trend holds on CIFAR100-C and ImageNet-C as below, showing that the gain comes from domain-invariant learning rather than the adapter. We will make this comparison explicit in the revision.
> | | CIFAR10-C | CIFAR100-C | ImageNet-C |
> |:---:|:---:|:---:|:---:|
> | Source | 28.2 | 35.4 | 55.8 |
> | Adapter + self-training | 18.0 | 27.0 | 45.0 |
> | TestDG | **7.7** | **23.3** | **42.5** |
>
> ___
> ### **4. Robustness to wrong domain-change detection**
>
> TestDG detects domain changes using a confidence threshold (Sec. 3.1), and Figure 5(a) shows that it is insensitive to this threshold. If domain changes are detected more frequently than actual changes, TestDG still functions effectively by detecting intra-domain shifts within a single domain and learning invariant features accordingly. Conversely, when a change is undetected, the shift is too small to exceed the threshold, so there is little domain gap to reduce. Following your suggestion, we further compare our confidence-based detection with *oracle boundaries*, and with *no detection* that never detects domain changes and lets the first-domain prototypes be used throughout.  As shown below, confidence-based detection performs on par with the oracle, implying that TestDG works well under imperfect boundaries, while removing domain-change detection increases the error, confirming that the detection still contributes. We will include these results in the revision.
>
> | Detection | Mean error |
> |:---:|:---:|
> | Confidence (ours) | **7.7** |
> | Oracle | 7.6 |
> | No detection | 10.3 |
>
> ___
> ### **5. Robustness under class imbalance**
>
> Thank you for the valuable suggestion. We evaluated TestDG on class-imbalanced test streams, where we resample each corruption so that 3 of 10 classes account for 90% of the samples and the remaining 7 share the other 10%. Since TestDG relies on pseudo-labels, class imbalance biases them toward the majority classes, which is an inherent limitation of pseudo-label-based adaptation. Nevertheless, as shown below, TestDG still shows a low error without collapse, while the error of the frozen source model stays high. We will note this limitation in the revision.
>
> | | Balanced | Imbalanced |
> |:---:|:---:|:---:|
> | Source | 28.2 | 29.3 |
> | TestDG | 7.7 |9.1 |
>
> ___
> ### **6. Using prototypes from multiple previous domains**
>
> Thank you for the great comment. As shown in the table below, using prototypes of more previous domains worsens both adaptation and generalization. During CTTA, the feature space is continually updated, so prototypes from earlier domains grow outdated and no longer represent their domains, and aligning to such stale references harms generalization. Retaining only the most recent domain thus performs best while keeping the memory constant regardless of the stream length. We will include this ablation in the revision.
>
> | Number of previous domains | CTTA | DG |
> |:---:|:---:|:---:|
> | 1 (ours) | **7.7** | **11.8** |
> | 3 | 8.3 | 12.7 |
> | 5 | 8.2 | 13.1 |

---

> ### Author Response · Authors · 2026-07-06
>
> ___
> ### **7. Random seeds and domain orders**
>
> Following your suggestion, we evaluated our method with 4 corruption orders and 3 random seeds. As shown below, the standard deviation is clearly below our 1.2%p improvement over the previous state of the art, showing that the gain is stable. We will report these statistics in the revision.
> | Setting | Mean error |
> |:---:|:---:|
> | 4 corruption orders | 7.87 ± 0.40 |
> | 3 random seeds | 7.81 ± 0.13 |
>
> ___
> ### **8. The "best" claim on ImageNet-C**
>
> Thank you for the comment. On ImageNet-C, TestDG and Continual-MAE perform on par at 42.5%, as the gaps among methods naturally shrink when the number of classes is large. We will revise the expression to "on par with the best (Continual-MAE)" in the revision.
>
> ___
> ### **9 and 10. Computational settings and cost**
>
>
> All numbers were measured on CIFAR10-C with a batch size of 40 on a single NVIDIA RTX A6000, under the same conditions for every method and with one gradient step per batch. CoTTA and ViDA average the teacher over many augmented views, which dominates their cost, whereas TestDG uses a single augmented view without averaging. As shown below, TestDG runs 4 forward passes per step and requires the fewest FLOPs and the lowest latency per step at comparable GPU memory. We will include these results in the revision.
>
>
> | Method | Forwards/step | TFLOPs/step | Latency (sec/item) | GPU memory |
> |:---:|:---:|:---:|:---:|:---:|
> | CoTTA | 35 | 82.20 | 6.8 | 47.3 GB |
> | ViDA | 12 | 31.10 | 6.5 | 33.7 GB |
> | Continual-MAE | 12 | 31.10 | 5.5 | 36.5 GB |
> | TestDG | **4** | **13.33** | **4.2** | 34.4 GB |
>
> ___
> ### **11. Preservation of class information**
>
> Thank you for the valuable suggestion. To verify that domain-invariant learning does not erase class information, we measure how well the classification features are grouped by class, before and after adaptation, using the source model as reference. As shown below, the intra-class distance shrinks much more than the inter-class distance, so both the inter/intra ratio and the silhouette score are improved substantially. The inter-class distance also decreases, but only because the whole feature space becomes tighter, and its ratio to the intra-class distance still grows. These results indicate that domain-invariant learning sharpens the per-class structure rather than erasing it. We will include these results in the revision.
>
> | Metric | Source | TestDG |
> |:---:|:---:|:---:|
> | Intra-class distance | 73.65 | **16.71** |
> | Inter-class distance | 36.72 | 25.30 |
> | Ratio (inter/intra) | 0.50 | **1.51** |
> | Silhouette score | 0.011 | **0.257** |
>
> ___
> ### **12. Generalization beyond held-out corruptions**
>
> Thank you for the comment. Since the generalization in Table 4 is evaluated only on held-out corruptions of CIFAR10-C, we further conduct the leave-one-domain-out evaluation on the ACDC benchmark, whose domains are adverse weather conditions in the real world. TestDG achieved the best generalization (mIoU) on these unseen real-world domains, indicating that its domain-invariant features generalize beyond synthetic corruptions. We will include these results in the revision.
>
> | Source | CoTTA | ViDA | TestDG |
> |:---:|:---:|:---:|:---:|
> | 58.3 | 58.5 | 58.4 | **59.0** |